# Analysis of the impact of expressway construction on soil moisture in road areas

**Yongyi Li**, **Zhan Xiao**, **Zhihao Li**, **Zexuan Jiao**, **Xingli Jia***

Road college, Chang'an University, Xi'an, Shannxi, China

* muyuyoo@163.com

## Abstract

To reveal the effect pattern of expressways on regional soil moisture, in this study, using trend analysis and buffer zone analysis methods, the data of VSWI (vegetation supply water index) in central Zhejiang Province from 2005 to 2016 were extracted from landsat7 satellite data using a single window algorithm, and spatial analysis was used to investigate the law of its differentiation. The results show that the multi-year average is 0.01879, between 0.01035–0.02774, showing a gentle decreasing trend, and there are obvious regional variations in space. We found that the impact of the new expressway and interchange on the VSWI in the buffer zone lasted for more than two years, and the VSWI increased in space farther away from the road, and this trend returned to normal at 8 km. Finally, the development patterns of the VSWI in the buffer zone of the newly established expressway and the interchange are approximately the same.

## 1. Introduction

Soil moisture, as a major environmental factor, is an important factor in evaluating the impact of expressways on the road environment. The construction process of expressways will certainly disturb the soil moisture of the road environment, resulting in changes, thus producing the corresponding distribution change law, that is, the spatial and temporal impact of expressway construction on the soil moisture of the road. Influence of expressway construction on road area soil moisture. Especially in areas with high vegetation cover, the impact of expressway construction on soil moisture in the road area is more obvious, and these areas are highly sensitive to changes in the road environment.

In this study, soil moisture was analyzed using satellite remote sensing image technology. Compared with traditional soil moisture monitoring methods, remote sensing technology has many irreplaceable advantages for monitoring soil moisture, as it can detect soil moisture around expressways in the region quickly, in real time, for a long period of time, dynamically, and with good spatial resolution in time [1]. Remote sensing monitoring of soil moisture can be divided into two main categories based on this principle: one category is based on the change in soil moisture causing a change in soil spectral reflectance. The other category is based on changes in plant physiological processes caused by drought, which change the spectral properties of leaves and significantly affect the spectral reflectance of the plant canopy. These can be divided into ground remote sensing, aerial remote sensing, and satellite remote

**Data Availability Statement:** All relevant data are within the paper and its Supporting Information files.

**Funding:** This study was funded by the National Key Research and Development Program of China

(grant numbers 2021YFB2600403, 2020YFC1512003), and by the Fundamental Research Funds for the Central Universities, CHD (grant number 300102212203), all awarded to YL. The funders had no role in study design, data collection and analysis, decision to publish, or preparation of the manuscript.

**Competing interests:** The authors have declared that no competing interests exist.

sensing. Remote sensing bands can be divided into visible and near-infrared remote sensing [2], thermal infrared remote sensing [3, 4], microwave remote sensing [5], microwave remote sensing monitoring methods [6], thermal inertia methods [7], vegetation supply water index (VSWI) methods [8, 9], and thermal infrared remote sensing monitoring methods [10]. In comparison, the vegetation supply water index method is a more mature and simple method for remote sensing inversion of soil water content using near-infrared and thermal infrared wavelengths and is one of the most commonly used methods for remote sensing inversion of soil water content [11–13].

Based on this approach, many scholars have further investigated soil moisture inversion: Elnaz Neinavaz et al. showed the unique advantages and greater accuracy of thermal infrared remote sensing techniques for monitoring terrestrial vegetation (e.g., vegetation water stress and inversion of biophysical parameters) [4]; Carlson et al. combined the soil-vegetation-atmosphere transport (SVAT) model with satellite-acquired surface radiation temperature and normalized vegetation difference index (NDVI) to determine surface soil water effectiveness and proposed the calculation of vegetation supply water index (VSWI) [14], and McVicar et al. further applied the calculation of the vegetation supply water index (VSWI) to land drought assessment; [15] Yun Liang, Shuobo Chen et al., based on MODIS data and actual soil water content measurements, fitted a good linear relationship between the vegetation supply water index (VSWI) and soil water content, which proved the feasibility of using the MODIS vegetation supply water index to monitor soil moisture [16, 17]. Yang L. et al. used mono-temporal FY-ID/AVHRR data to confirm the applicability of the vegetation supply water index method to large-scale drought monitoring; [18] Lidiane CristinaCosta et al. used the vegetation supply water index (VSWI) method to analyze the relationship between terrestrial drought and agricultural land change, and showed that VSWI can effectively identify soil water content and its restoration effects [19].

As a product of infrastructure construction, the construction of highways is bound to have an impact on VSWI. Since roads are ribbon structures, their construction also affects the area along the road [20, 21], and the road itself affects the vegetation and soil along the road [22, 23]. At present, most studies only discuss the surface temperature of the road network in urban areas [24], however, highways, as an important part of the development of social infrastructure, should pay attention to sustainable development during the sustainable development of roads [25, 26] and conduct in-depth research and analysis based on their own changes, and if the environmental disturbance of road construction cannot be analyzed, the sustainable environmental development along the road cannot be development is assessed, which in turn affects road network planning. In this paper, in order to discuss the changes of VSWI along the road domain caused by the continuous construction of the road and its impact on the surrounding environment, internal analysis of soil moisture in the regional road network area is conducted from the perspective of soil moisture along the road, using Landsat 7 remote sensing image data to analyze the evolution of the influence mechanism of soil moisture along the road in the process of continuous changes, and also to explore the influence factors of the road itself, The research process is shown in Fig 1. It aims to guide the future road network encryption planning process in terms of VSWI and contribute to environmental protection, which is important for the sustainable and healthy development of roads.

## 2. Materials and methods

### 2.1. Study area

The study area is located in central Zhejiang Province, in the Yangtze River Delta region, with coordinates of 119˚14′-120˚4630' E and 28˚32′-29˚41N. It is located in the eastern part of

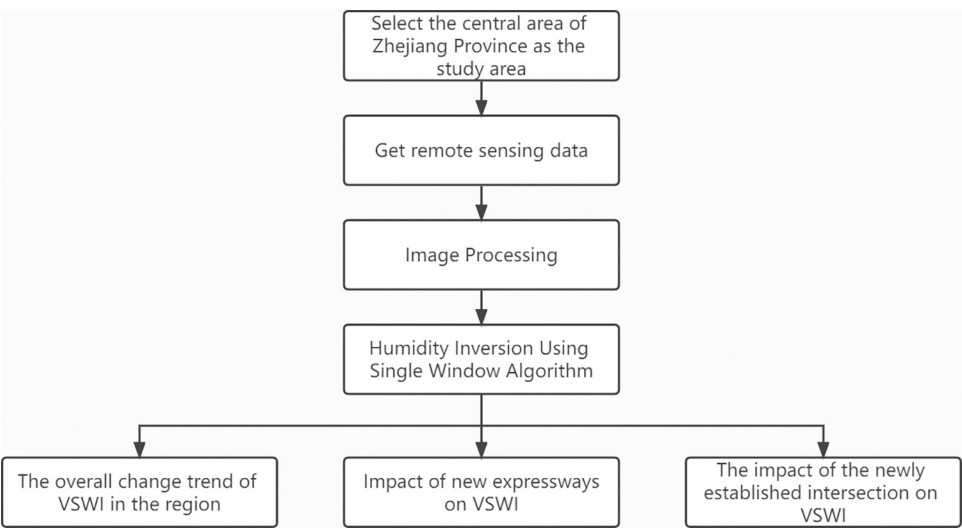

**Fig 1. Research process.**

Jinqu Basin, a hilly basin area in central Zhejiang Province, with high topography from north to south and low topography in the middle, and the vegetation type belongs to the subtropical evergreen broad-leaved forest zone, which is sensitive to the changes of soil moisture. In this study, expressways with a large construction scale and long construction time in the past 12 years in the study area were selected as the research objects, namely the Yongjin, Jinliwin, and Dongyong expressways, whose construction and completion times and lengths are shown in Table 1. The locations of the study area and the expressways are shown in Fig 2.

## 2.2. Data sources

The geographic remote sensing data in this study were obtained from Landsat7 data products with a spatial resolution of 30m and temporal resolution of annual (total annual cycle), spanning the period of 2005–2016, and Remote sensing data of September of each year in 12 years with the same amount of clouds and the same region were selected. The data are remotely sensed data, and the initial satellite remote sensing data are first processed into vegetation cover, radiation brightness, temperature, and humidity effects by atmospheric correction and radiation calibration using the Environment for Visualizing Images(ENVI) remote sensing data processing platform. Then, the abnormal values of the processed image data were removed, and the vegetation supply water index (VSWI) was processed by unit conversion using ArcGIS technical software. Finally, the vegetation supply water index (VSWI) values of 12 years were cropped with the boundary of the study area as a mask to obtain the vegetation supply water index (VSWI) for a total of 12 years from 2005 to 2016. The average value of vegetation supply water index (VSWI) represents the average value of each image element attribute in the region.

**Table 1. Construction and completion time and length of each expressway.**

| Expressway | Construction time/year | Completion time/year |
|---|---|---|
| Yongjin | 2003 | 2005 |
| Jinliwin | 2008 | 2009 |
| Dongyong | 2009 | 2015 |

 

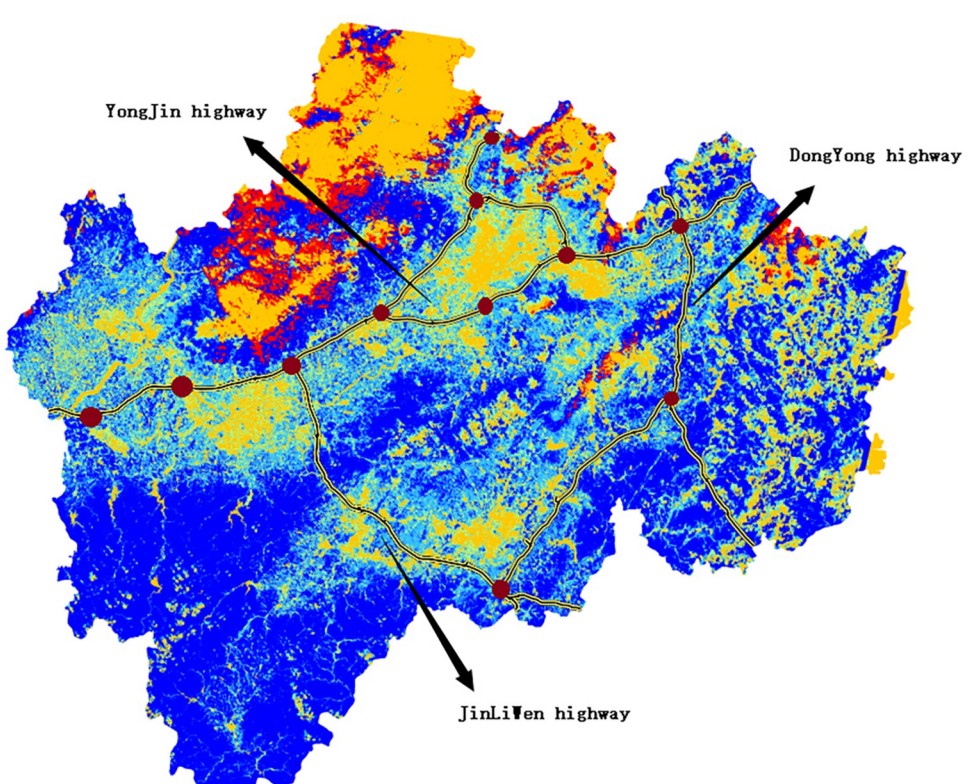

**Fig 2. Schematic diagram of the study area.**

Road data were plotted using the Geodata Spatial Cloud Landsat7 ETM SLC-off satellite digital products (2005–2016) and OpenStreetMap as references. To avoid the inconsistency of the geographic and projection coordinates of various spatial data, all spatial data were converted into a unified coordinate system with the geographic coordinate system as GCS_WGS_1984 and the projection coordinate system as WGS_1984_UTM_Zone_47N.

## 2.3. Data processing methods

**2.3.1. Data pre-processing.** When using remote-sensing images for analysis, clear and standard remote-sensing images are required to prevent excessive errors. However, there are obvious banding effects on the remote sensing images of MODIS data; therefore, data pre-processing is performed on the remote sensing images. Because the main cause of banding is not noise, it cannot be solved simply using the filtering method in signal processing. Considering the need for spectral fidelity, the G-S fusion method was chosen to deband Landsat Landsat7/ETM+ images [27]. Geometric correction, atmospheric correction, de-clouding, and radiometric calibration were performed sequentially on the processed images to obtain the images required for the model.

**2.3.2. Vegetation supply water index model.** The VSWI was calculated from Moderate Resolution Imaging Spectroradiometer (MODIS) remote sensing data, normalized difference vegetation index (NDVI), and land surface temperature (LST) on Terra (EOS AM) and Aqua (EOS PM) satellites [14, 28, 29]. The index indicates that vegetation is in a drought state when NDVI values are low (low photosynthetic activity) and vegetation temperature is high (water stress) [30–32]. Therefore, this index can indirectly reflect soil water content, as shown by the

fact that the lower the VSWI value of the crop when affected by drought, the lower the soil water content of the crop, and vice versa.

$$VSWI = NDVI/Ts \tag{1}$$

VSWI is the ratio of NDVI to Ts (i.e., land surface temperature LST), where Ts is the surface temperature; NDVI is the normalized difference vegetation index.

### 2.3.3. Calculation of relevant parameters.

*1. NDVI.* NDVI is the best indicator of vegetation cover and growth status, and the normalization of NDVI reduces the effect of instrument calibration errors on single bands as well as the effect of surface two-way reflections and atmospheric effects. In [33, 34], the NDVI was calculated from the data after cloud removal, and the maximum value composite (MVC) method was used to select the maximum value of the NDVI corresponding to each point and composite an image to remove the influence of clouds as much as possible [35, 36]. The specific formula for calculating NDVI is as follows:

$$NDVI = (NIR-R)/(NIR + R) \tag{2}$$

NIR is the reflection value in the near-infrared band and R is the reflection value in the red band. It can be calculated directly using the band math tool in ENVI5.3 software.

*2. $T_s$.* Surface temperature is often defined as the skin temperature of the Earth's surface (skin temperature). The heterogeneity of the ground surface leads to the complexity of the remote sensing inversion of surface temperature. For a densely vegetated surface, the remote sensing inversion of surface temperature refers to the surface temperature of the vegetation leaf canopy [37]. For a sparse surface, the surface temperature is the mixed average of the temperatures of the vegetation leaf canopy, ground surface, etc. [38, 39]. There are three main types of surface temperature inversion based on remote sensing: the radiative transfer equation method [40], single-window algorithm [41], and split-window algorithm [42]. In this study, a single-window algorithm was used for inversion.

$$T_s = \{a(1-C-D) + [b(1-C-D) + C + D]T_b-DT_a\}/C \tag{3}$$

$$C = \varepsilon \times \tau \tag{4}$$

$$D = (1-\tau)[1 + (1 - \varepsilon)\tau] \tag{5}$$

$$T_a = 16.0110 + 0.92621T_0 \tag{6}$$

$T_a$ is the average atmospheric action temperature (K), $T_b$ is the brightness temperature (K), $T_s$ is the inverse temperature (K), a and b are the correlation coefficients, -67.35535 and 0.45861 when the temperature is between 0 and 70°C, respectively, C and D are intermediate quantities, $T_0$ is the near-surface temperature in K, $\tau$ is the atmospheric transmittance, and $\varepsilon$ is the surface-specific emissivity.

## 2.4. Research methodology

### 2.4.1. Trend analysis.
Trend analysis is a method of fitting regressions to time-varying data to synthetically simulate and predict the evolution of regional patterns in a certain time series through the spatial variation characteristics of individual image elements in different periods. In this study, the trend analysis method was used to analyze the spatial trends of VSWI, and the slope of the regression trend indicates the trend of each raster point variable

during the study years. The calculation formula is specified as follows [43]:

$$\theta = \frac{n \times \sum_{i=1}^{n}(i \times VSWI_i) - \sum_{i=1}^{n} i \sum_{i=1}^{n} VSWI_i}{n \times \sum_{i=1}^{n} i^2 - \left(\sum_{i=1}^{n} i\right)^2} \qquad (7)$$

where θ is the slope of the regression trend, and its value is positive when it indicates an increasing trend of VSWI during the study period, and vice versa; n is the time span, which is 12 in this study; and i = 1, 2, . . ., 12 corresponds to 2005, 2006, . . ., 2016, respectively, denoting the VSWI value in year i.

**2.4.2. Stability analysis.** The relative degree of fluctuation of the data over time can be reflected by the coefficient of variation in the stability analysis, which was used in this study to analyze the stability of VSWI changes, and the formula is shown below [44].

$$C_v = \frac{\sigma}{\bar{\chi}} \qquad (8)$$

where x̄ is the multiyear mean VSWI. σ is the standard deviation. $C_v$ The smaller the value, the more concentrated the data distribution, the smaller the fluctuation in the data over time, and the better the stability of the data. In contrast, the data distribution was more discrete, and the data fluctuated significantly with time.

**2.4.3. Road buffer zone analysis.** A buffer zone of 200m, 500m, 1 km, 2 km, and 5 km was made to both sides of the expressway for a total of five scales for comparison, and then the comparison was made in different expressways. The construction of the expressway led to a significant change in land cover within the road area, which directly affected the soil moisture value within the road area. In this study, we used the trend and spatial analysis methods of Geographic Information System (GIS) to analyze the spatial and temporal effects of expressway construction on the vegetation supply water index (VSWI) in the Zhejiang Jinhua region for 12 years from 2005 to 2016. The established multiloop buffer area is illustrated in Fig 3.

Twelve interchange nodes were constructed on three expressways (Yongjin Expressway, Dongyong Expressway, and Jinliwin Expressway section) in Jinhua from 2005 to 2016, which were numbered and analyzed for their buffer zones with radii of 200m, 500m, 1, 2, and 5 km, respectively. As shown in Fig 4, nodes 1, 2, 4, 5, 6, 7, and 8 are located on the Yongjin Expressway; nodes 3, 10, and 11 are located on the Jinliwen Expressway section; and nodes 9 and 12 are located on the Dongyong Expressway. In this study, the spatial analysis method of Geographic Information System (GIS) was used to analyze the spatial and temporal effects of the construction of interchanges on the VSWI in the Jinhua region of Zhejiang Province for twelve years from 2005 to 2016.

# 3. Results and discussion

## 3.1. Overall diagram of vegetation supply water index in the study area

By analyzing and calculating the remote sensing images of the study area through the above data processing methods, the overall schematic diagram of the vegetation supply water index (VSWI) in the study area for each year from 2005 to 2016 can be obtained, as shown in Fig 5. (Shown here is the VSWI schematic for 2015).

## 3.2. Overall spatial and temporal variation of vegetation supply water index (VSWI)

**3.2.1. Time variation.** The vegetation supply water index (VSWI) in the study area had a 12-year average value of 0.01879, ranging from 0.01035 to 0.02774, with the lowest VSWI

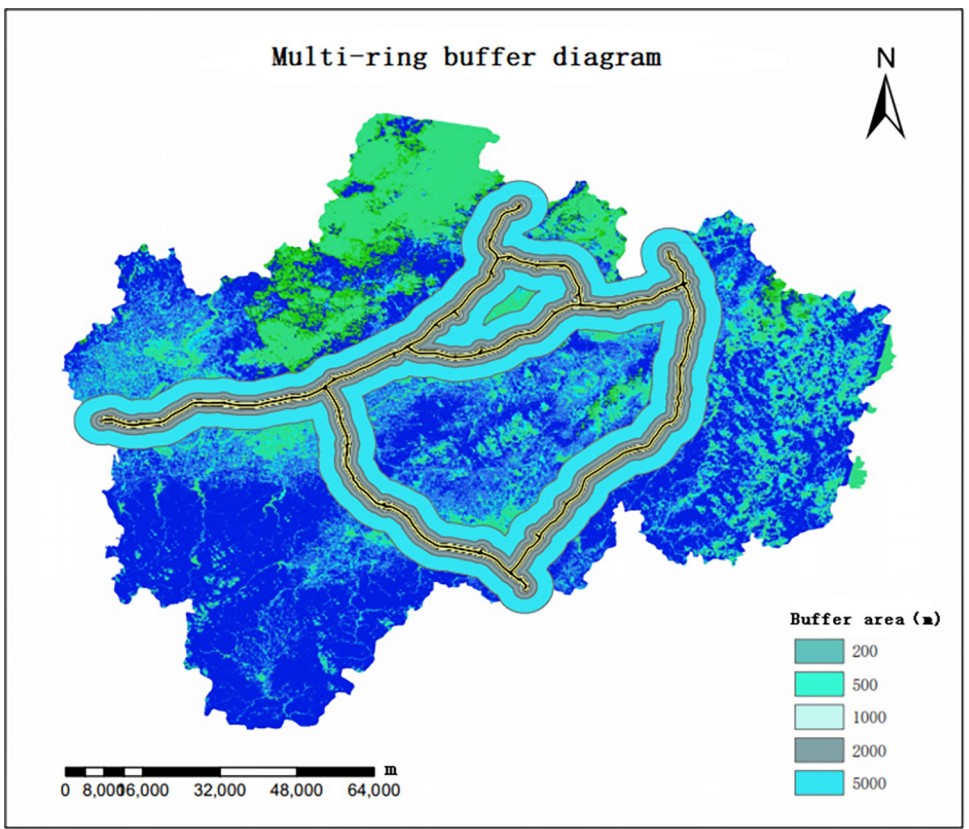

**Fig 3. Diagram of buffer area.**

value in the 12-year period occurring in 2011 and the highest value in 2013. In terms of the trend (Fig 6), from 2005 to 2011, there was a decreasing-increasing-decreasing trend, and the lowest annual average VSWI value in the 12 years was reached in 2011. After 2011, there was an increasing trend, and the highest annual average VSWI value was reached in 2013, after which there was a decreasing-increasing-decreasing trend. Overall, the annual average VSWI has certain volatility, which indicates that the humidity in central Zhejiang changes significantly. The trend line of annual average variation shows that the annual average VSWI has a flat and decreasing trend.

**3.2.2. Spatial variation.** The spatial distribution of the multi-year mean of the VSWI in the study area from 2005 to 2016 is shown in Fig 7, showing a clear spatial distribution characterized by a mean value of 0.01879 and an overall clear variability of regional variability. To better demonstrate the state of soil moisture differences within the study area, the natural breakpoint method was used in the Geographic Information System (GIS), i.e., the optimal arrangement of values in groups was determined by iteratively comparing the sum of squared differences between the mean value and the observed value for each group and element in the group, and the optimal classification calculated from this could determine the breakpoint of values in the ordered distribution to minimize the sum of squared differences within the group. The fitted interrupted values of p obtained according to the natural breakpoint method are 0.0029, 0.0138, 0.0247, and 0.0356. Therefore, the regions are graded according to VSWI as shown in Table 2, defining regions with $p < 0.0029$ as primary regions, regions with $0.0029 < p < 0.0138$ as secondary regions, and regions with $0.0138 < p < 0.0247$ as tertiary

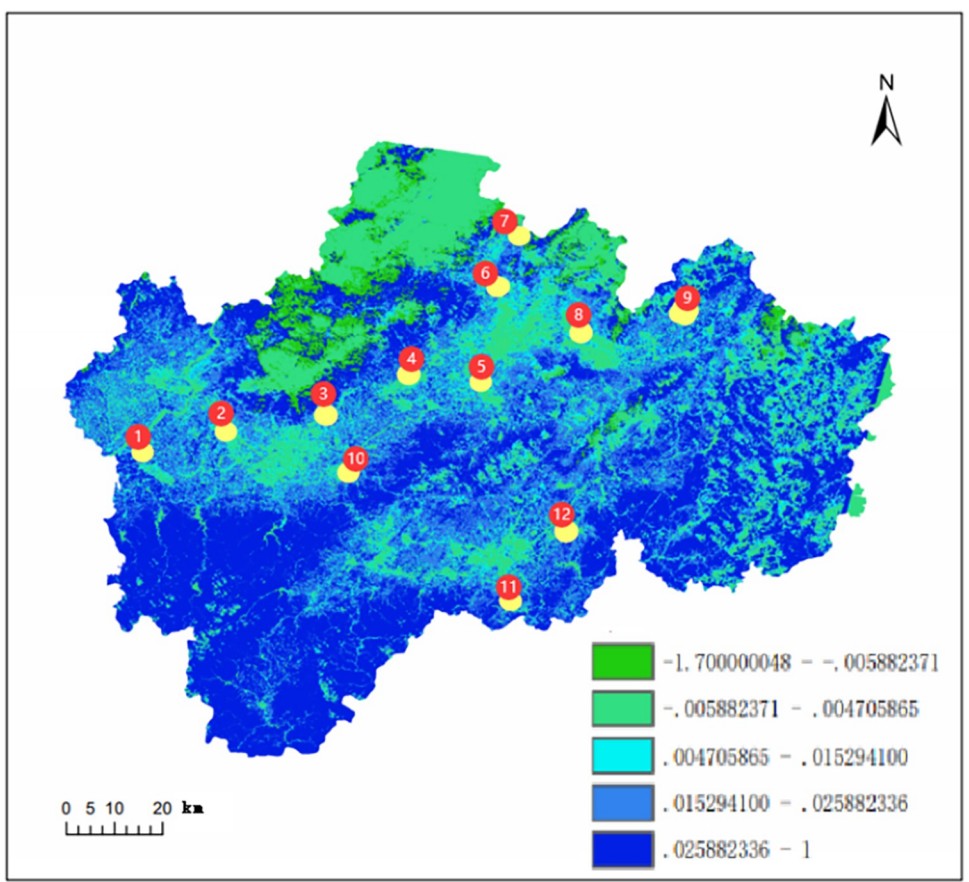

**Fig 4. Schematic diagram of the interchange node.**

regions, $0.0247 < p < 0.0356$ as quaternary regions, and $p > 0.0356$ as quintuple regions, with higher grades indicating higher soil moisture.

The area of Class I in the study area was 13.36%, the area of Class II was 10.24%, the area of Class III was 63.7%, and the area of Class IV was 10.77% of the total area, while the area of Class V was only 1.93%.

Based on the topography of the study area and the VSWI classification it can be seen that, most of the central and western areas of the study area are urban plain areas, and there is a staggered road network on the surface. The hills with higher altitude are located in the fourth-level area; the situation in the north, south and east of the study area is the opposite, there are only a few roads for passage, most of the terrain is mountainous area, in the third-level and fourth-level areas; At the same time, the three new expressways are located in the first- and second-level areas.

The overall spatial variation trend of the overall regression trend slope change of VSWI in the study area was calculated by trend analysis, as shown in Fig 8. This paper is based on the idea of trend analysis method, through the spatial change characteristics of individual image elements in different periods, the integrated simulation predicts the regional pattern evolution process of certain time series, using the weighted average of the absolute value of residuals as the minimization objective function, so it is not easy to be influenced by extreme values more robustly, and the influence of the explanatory variables on the explanatory variables can be shown on the whole distribution of the latter. When the slope of the regression trend has a

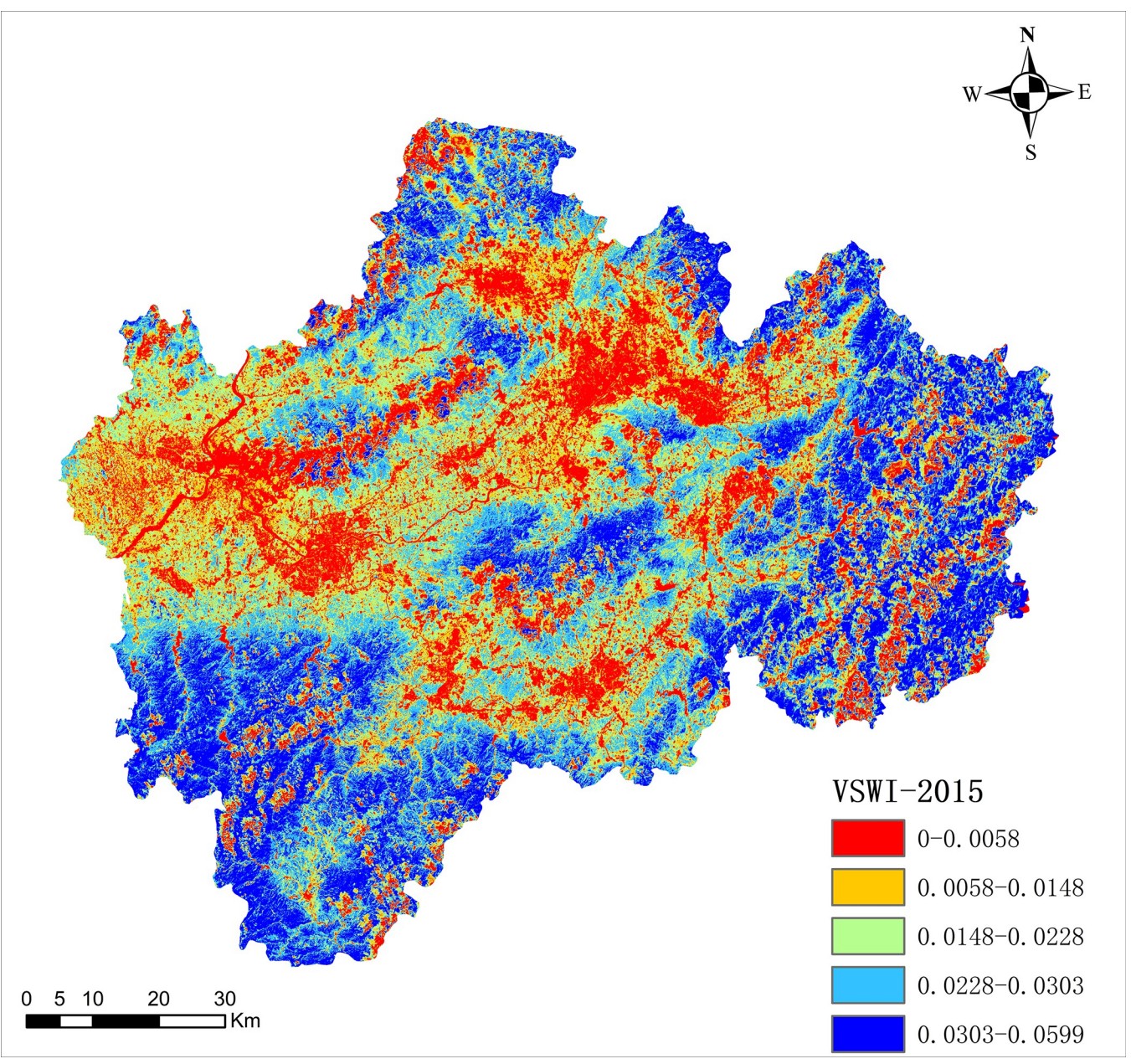

**Fig 5. Overall schematic of VSWI in 2015.**

positive value, it indicates that the VSWI has an increasing trend during the study period, and vice versa, a decreasing trend.

Among them, The area with a slope of the regression trend greater than 0 accounted for 44.56% during 2005–2016, and the remaining 55.44% of the area showed a negative growth trend. Areas with regression trend slopes less than 0 were distributed in the western and north-central parts of the country, and the VSWI showed a flat decreasing trend in general. It can be seen that the overall soil moisture in the study area showed a gentle decreasing trend over a 12-year period.

**3.2.3. Stability analysis.** The coefficient of variation of the VSWI values in the study area was obtained according to the stability calculation formula in Geographic Information System

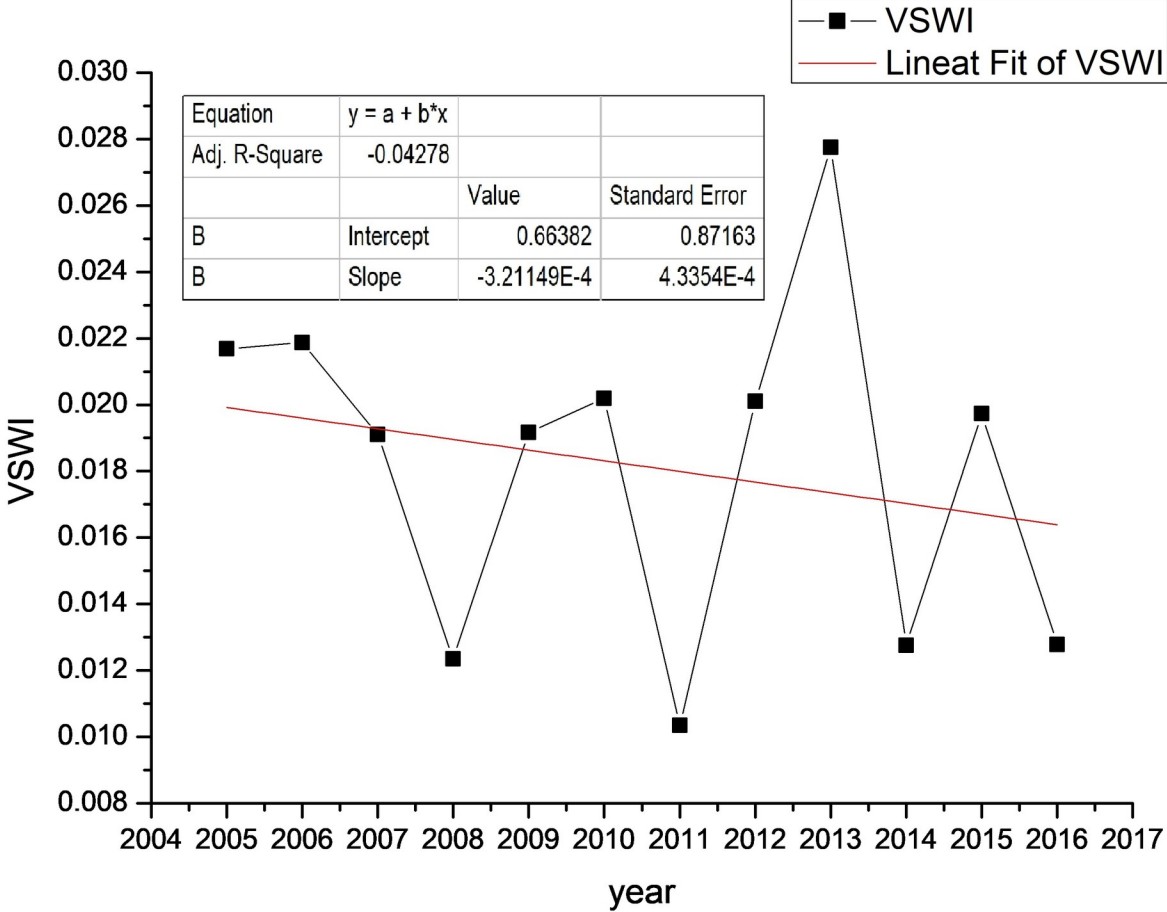

**Fig 6. Overall average VSWI by year in the study area.**

(GIS), and the results are shown in Fig 9. After removing the outliers through the filtering function, the variation coefficient of the VSWI variation coefficient in the study area from 2005 to 2016 is relatively stable. It can be seen that there is a new expressway in the middle of the study area, which produces changes in the VSWI values and the stability is affected, but the variation coefficient value is small and the data distribution is more concentrated, which indicates that the fluctuation of the data over time is small and has good stability.

### 3.3. Impact of new expressways on vegetation supply water index (VSWI)

**3.3.1. Time variation pattern.** The Dongyong expressway was completed in 2015, and it is known from the statistical data that the average value of the road area VSWI was 0.0158 when there was no project, and the VSWI decreased to 0.1298 after the completion of the expressway, which was approximately 17.8%, and there was basically no change in VSWI until 2016. The study area Jinli-Wen expressway section was completed in 2009, and from the statistical data, it can be seen that the average value of road area VSWI was 0.01547 when there was no project, and the average value of VSWI from 2009 to 2016 after the completion of the high-speed was 0.011913, a decrease of about 23%, showing a downward trend. The Yongjin expressway section in the study area was completed in 2006, and from the statistics, it can be seen that the average value of VSWI in the road area was 0.0157 when there was no project.

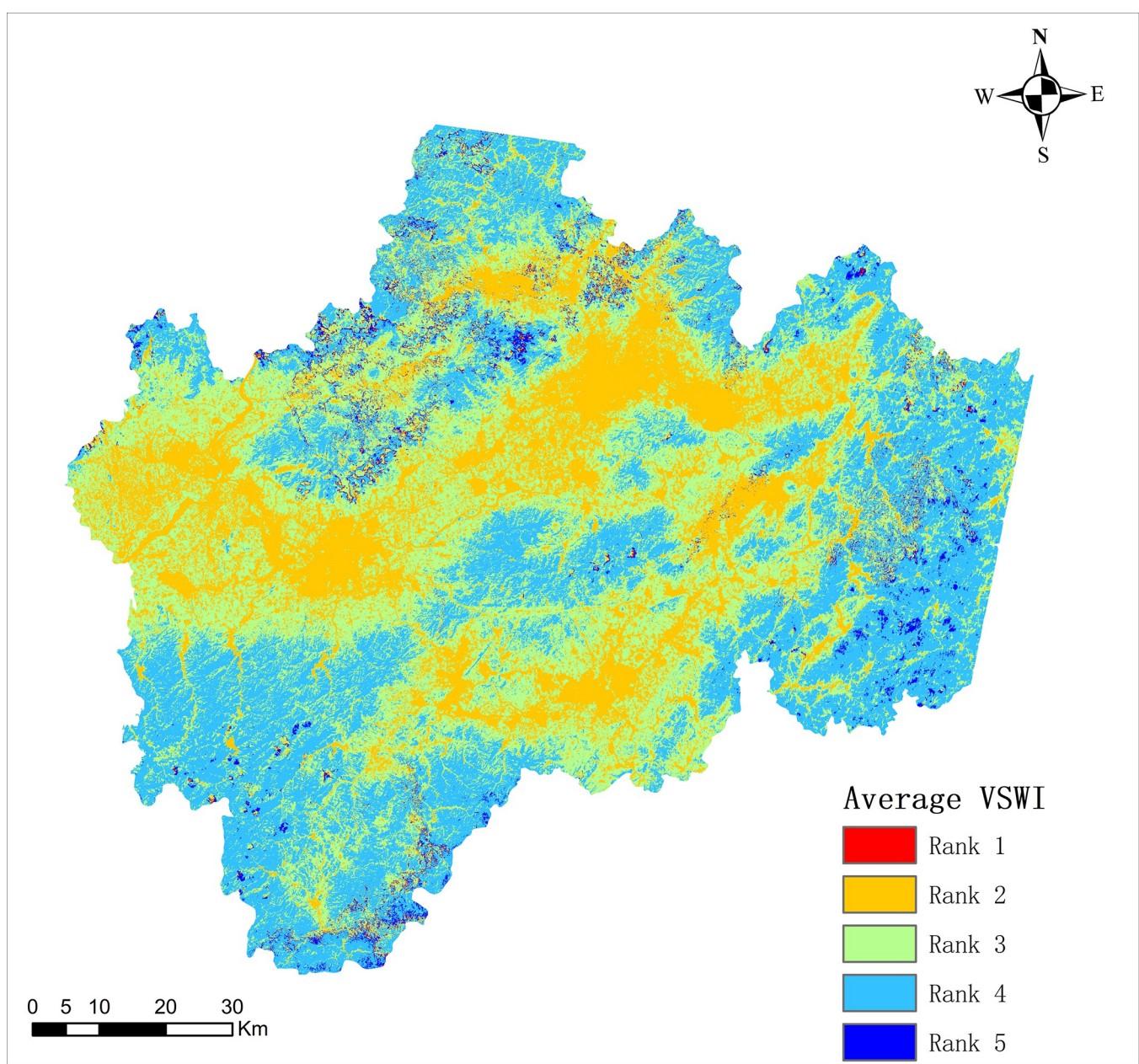

**Fig 7. Average VSWI from 2005–2016.**

**Table 2. Regional grading.**

| Rank | Discontinuous value p | Area ratio |
|------|-----------------------|------------|
| 1 | p<0.0029 | 13.36% |
| 2 | 0.0029<p<0.0138 | 10.24% |
| 3 | 0.0138<p<0.0247 | 63.7% |
| 4 | 0.0247<p<0.0356 | 10.77% |
| 5 | p>0.0356 | 1.93% |

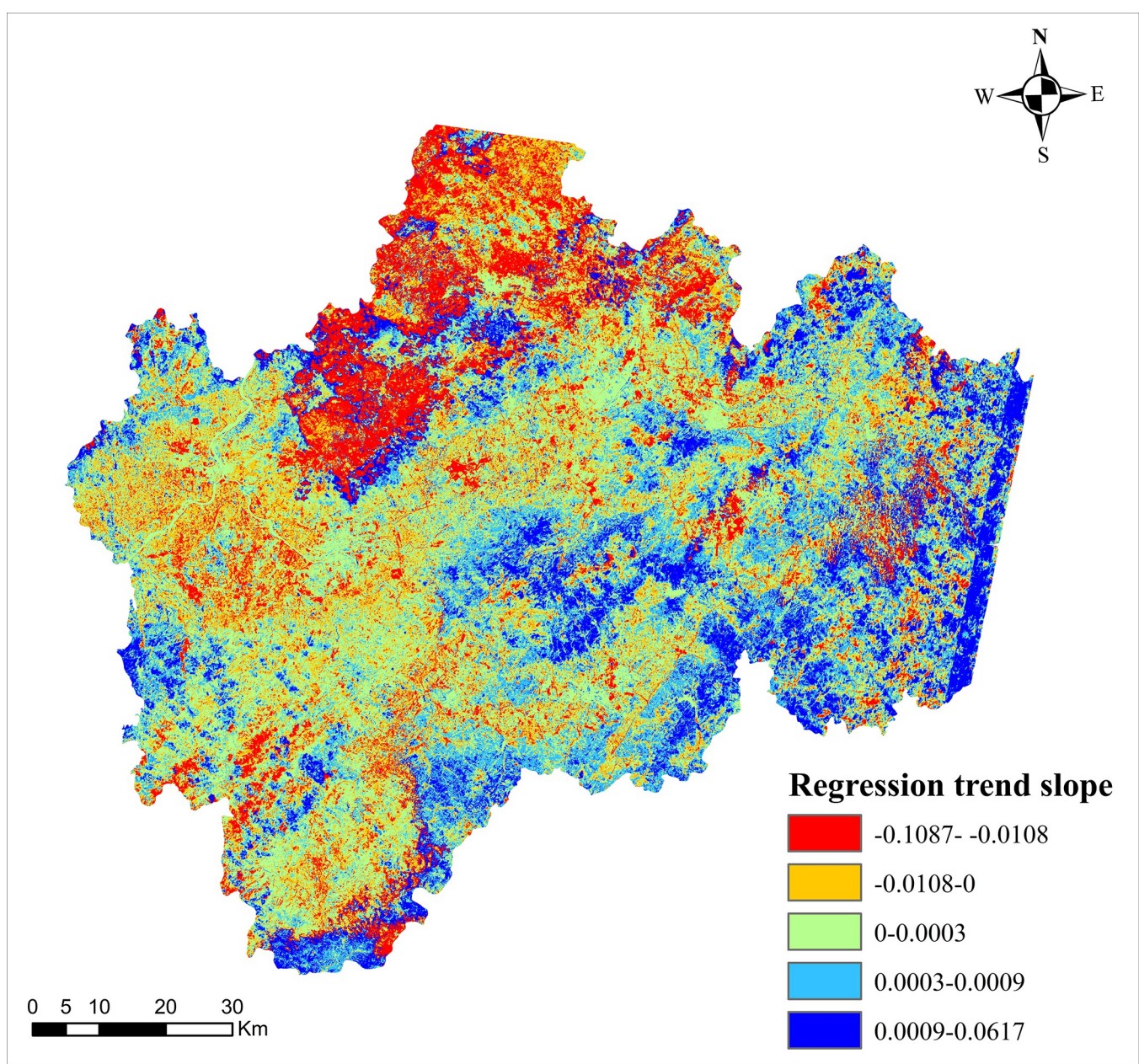

**Fig 8. General spatial trend of the overall regression trend slope of VSWI in the study area.**

Because of the impact of construction, the environment within a certain area of the road area changed and plummeted to 0.01051 in 2007, which eventually reduced the average value of VSWI from 2007–2016 reduced to 0.01011, a decrease of approximately 35.6%. Two years after construction, the VSWI showed a small increase and then continued to decline in a fluctuating manner. Meanwhile, the trend lines of VSWI averages of the three new expressways (as shown in Fig 10) can be seen to be in a downward trend. It can be seen that the new expressway has a certain impact on the road environment, and a few years after the completion of the new expressway, the VSWI has slightly improved but still has not returned to the level before the completion. After the completion of the expressway, the low VSWI in the road area

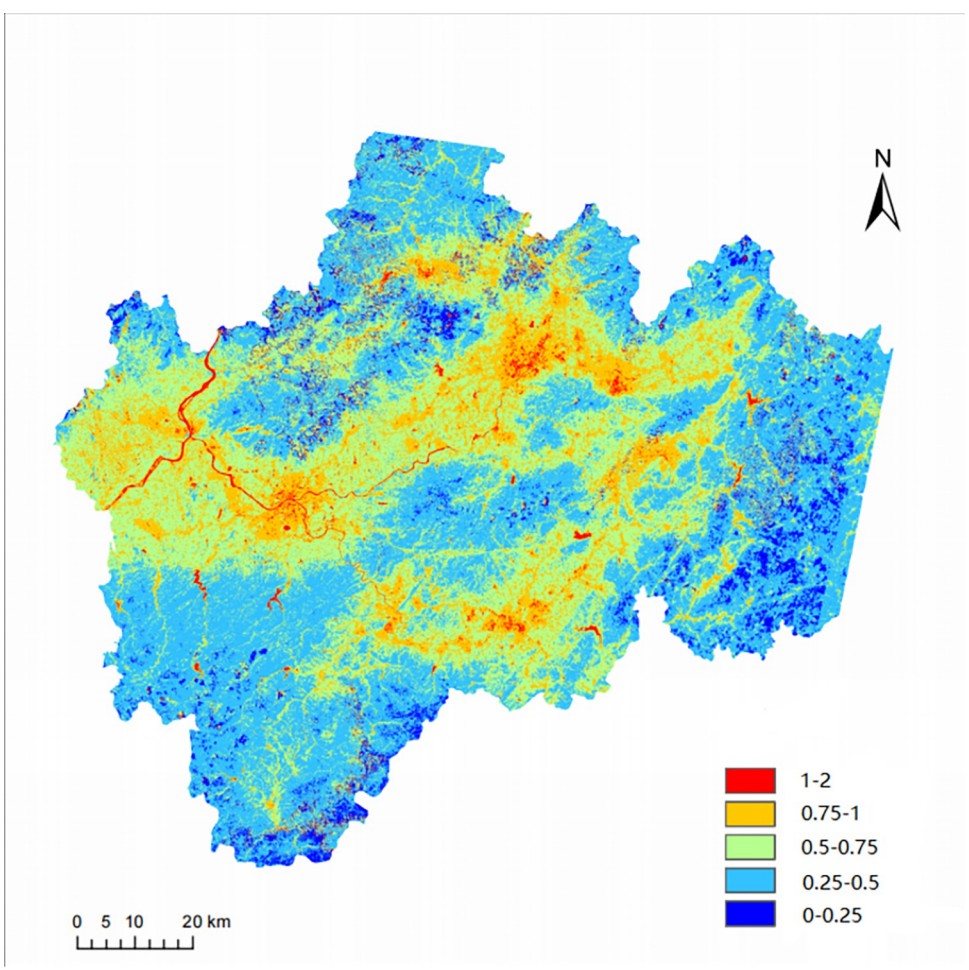

**Fig 9. Schematic diagram of coefficient of variation.**

was first caused by the land cover change caused by the new expressway and then produced a certain degree of irreversible damage to the environment.

Figs 11–13 show the time effect of the VSWI after the construction of the project by trend analysis. The spatial distribution of the Dongyong Expressway multi-year VSWI change trend is shown in Fig 11. Before 2009, the expressway had not been built yet, 51.44% of the area showed a decreasing trend, with an interannual change rate of 0.14551%; and from 2009 to 2015, 53.50% of the area showed a decreasing trend, with an interannual change rate of 0.09808%; and by the completion of the expressway construction in 2016, 97.39% of the area showed a decreasing trend, with an interannual change rate of -0.82906%. The spatial distribution of the Jinliwen Expressway multi-year VSWI change trend is shown in Fig 12. During the construction period from 2008 to 2009, 49.20% of the area showed a decreasing trend, with an interannual change rate of 0.55533%; from 2010 to 2016 after completion, 79.43% of the area showed a decreasing trend, with an interannual change rate of -0.14312%. The trend of VSWI in these years indicates that the humidity within the buffer zone shows a fluctuating change trend. 0.14551% to -0.82906% and 0.20121% to -0.14312%, from a positive rate of change to a negative rate of change, the regional VSWI is in a continuous decline, indicating that the construction of the expressway has directly changed the VSWI in a positive growth trend to a

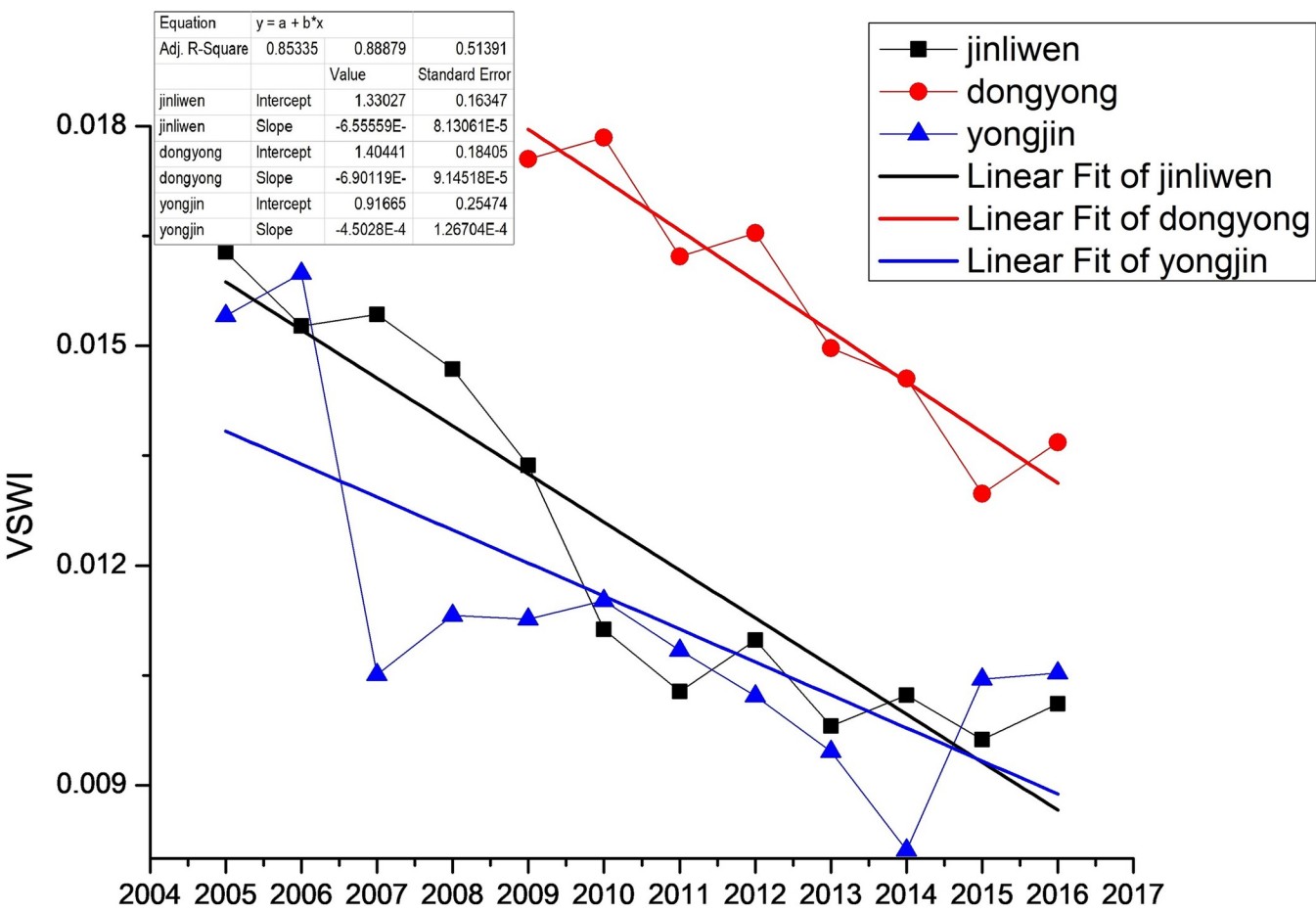

**Fig 10. Average value of VSWI in the 1km buffer zone of the three new expressways.**

negative growth state, that is, the impact of the new expressway on the VSWI in the buffer zone is very large. The spatial distribution of the Yongjin Expressway multi-year VSWI change trend is shown in Fig 13. From 2005 to 2007, 76.38% of the area showed a decreasing trend, and the average value of the interannual change rate was -0.2456%, indicating that the expressway construction phase for the road area VSWI showed negative growth. From 2007 to 2016, the decreasing area accounted for 38.38%, and the average value of the interannual change rate was -0.00957%, which is still in a decreasing trend. The large change in interannual change rate is due to the large time span, and the road area environment started to recover on its own. In general, the VSWI in the buffer zone continuously declines from planned construction to the completion of construction.

With the existing known VSWI years, the new expressway will have a significant impact on the roadway VSWI in time, which will last for at least two years.

**3.3.2. Spatial variation pattern.** Figs 14 – 16 shows the trend of the VSWI with buffer distance in different years for each expressway. Fig 14 shows the average value of the VSWI for different buffer distances on the Dongyong Expressway. For Dongyong Expressway, from the distance point of view, the VSWI of each year shows a significant increasing trend with the increase of road median distance, which is consistent with the theory of "the farther away from the road, the better VSWI trend"; from the year point of view, the completion of Dongyong

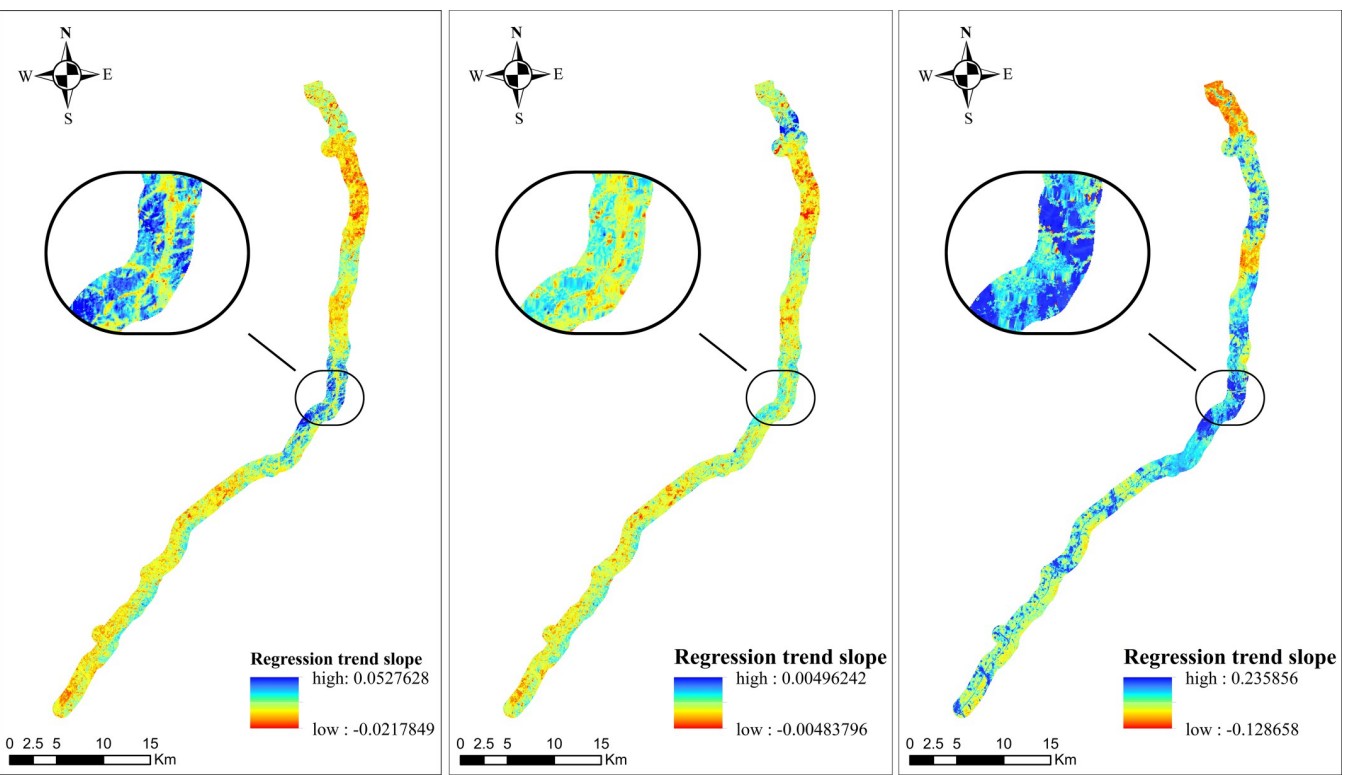

**Fig 11. Trend of multi-year VSWI spatial distribution in the buffer zone of Dongyong Expressway.** (**a**) Average trend of 2005-2008(last year) (**b**) Average trend of 2009–2014 (during construction) (**c**) Average trend of 2014-2016(finished).

Expressway in 2015 has a very big impact on the change of VSWI of the road area. The VSWI value continues to rise, and within 0.2–0.5 km, the VSWI increases the most, by 7.35%, compared to other buffer distances. One year after the construction of the project, there was a slight increase in the VSWI, but the increase in the VSWI within the buffer distance of different roads was more stable.

Fig 15 shows the average value of the VSWI for different buffer distances in the high-speed section of Jinliwine. From the distance point of view for the Jinliwine high-speed section, the VSWI of each year shows a significant increasing trend with the increase in the road median distance, with the largest increase from 1 to 2 km to 2–5 km, except for 2008; the VSWI of each buffer area of 2–5 km increased significantly compared with 1–2 km, indicating that the new Jinliwine high-speed section The expressway has the most obvious influence on its surrounding 2–5 km, and the average value of VSWI within 0–1 km basically remains stable, and the completion of the new expressway in 2010–2011 had a significant impact on the change of VSWI in the road area, followed by a fluctuating increase in 2012.

For the Yongjin Expressway, Fig 16 shows the average value of VSWI for different buffer distances of Yongjin Expressway. From the distance, VSWI shows a slow increasing trend with the increase in road median distance in each year, the change is not obvious in the range of 0.5–2 km, and VSWI shows a significant increasing trend with the distance after 2 km. The completion of the Yongjin Expressway in 2007 has a significant impact on the change of VSWI in the road area, and the value of VSWI decreases significantly. In 2007, the VSWI within 2–5 km increased by 10.27%, and the average value of the VSWI in each buffer area was the lowest in 2008.

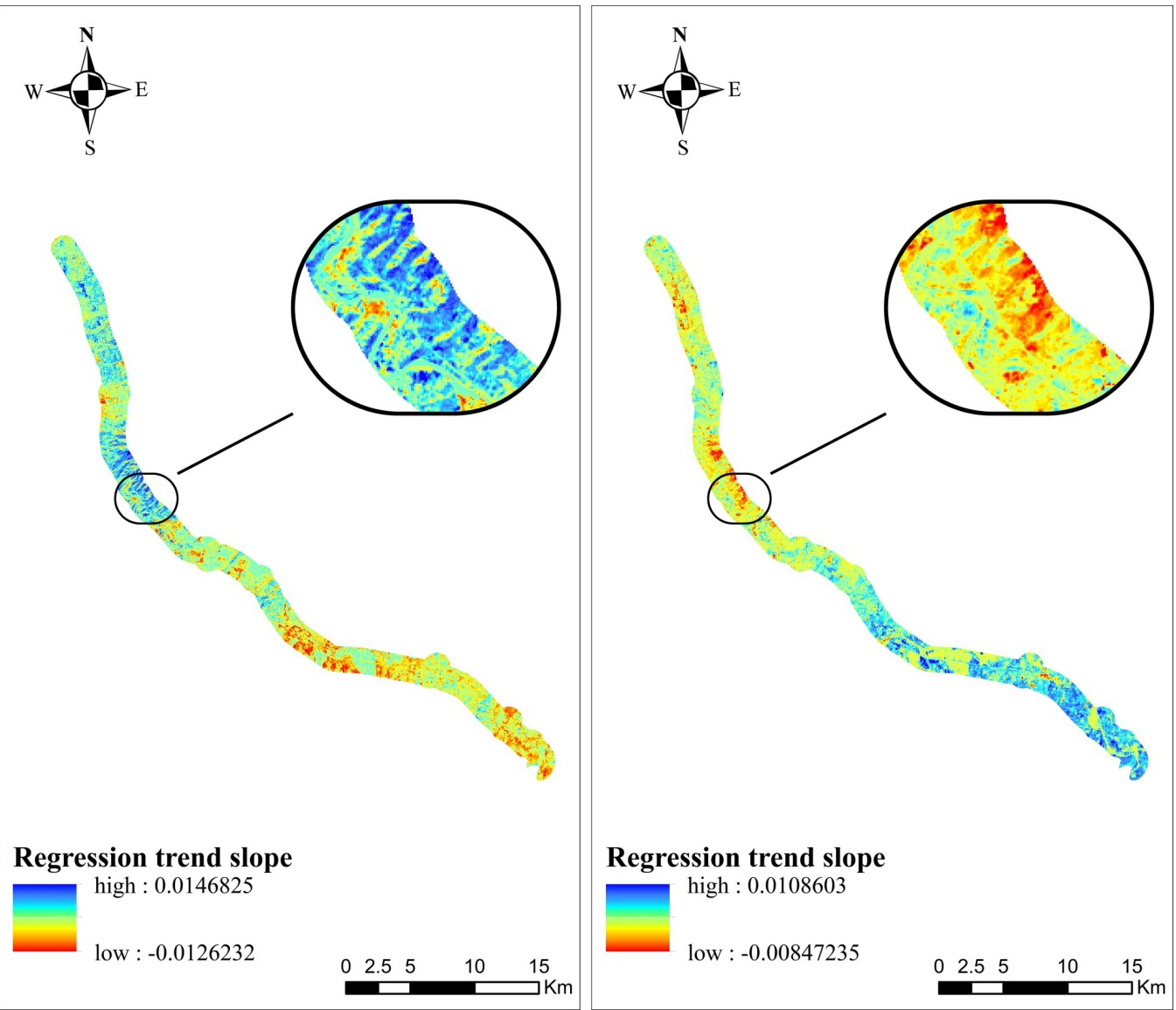

**Fig 12. Trend of multi-year VSWI spatial distribution in the buffer zone of Jinliwin Expressway.** (**a**) Average trend of 2008–2010 (during construction) (**b**) Average trend of 2010-2016(finished).

In general, the three new expressways show a spatial pattern of "the further away from the expressway the higher the VSWI". This means that the farther away from the road, the less the environment is affected by the new road and the lower the intensity of human activities or the ability to affect the environmental VSWI.

### 3.4. Effect of interchange nodes on vegetation supply water index (VSWI) in the study area

**3.4.1. Time variation pattern.** Table 3 shows the average values of the VSWI within 1 km for each interchange node before and after construction. It can be seen that the average value of VSWI of the interchange nodes with 1 km as the radius of radiation is in a decreasing state after the construction, among which the most obvious decrease is interchange node 9 on the

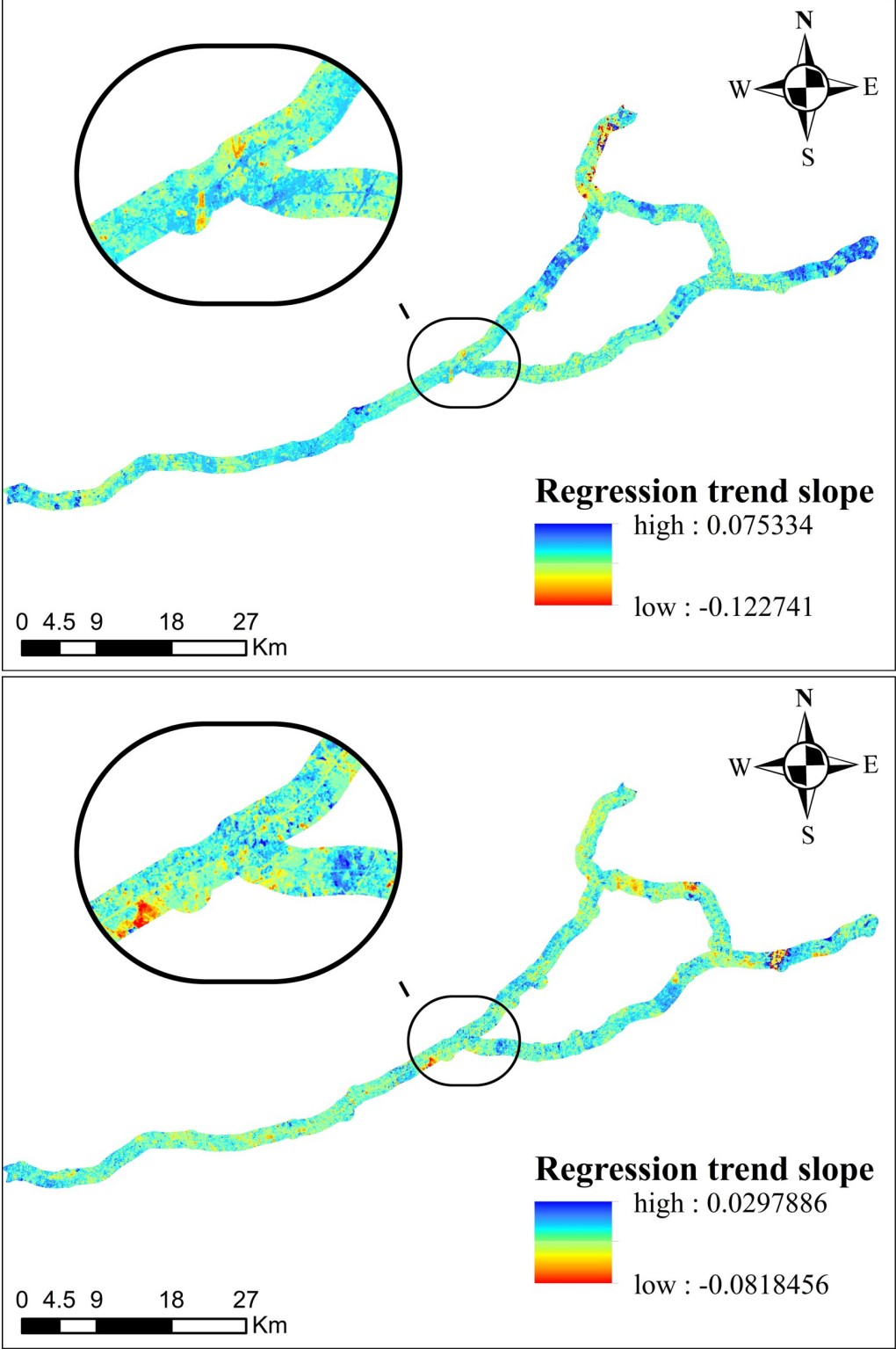

**Fig 13. Trend of multi-year VSWI spatial distribution in the buffer zone of Yongjin Expressway.** (**a**) Average trend of 2005-2008(during construction) (**b**) Average trend of 2008-2009(finished).

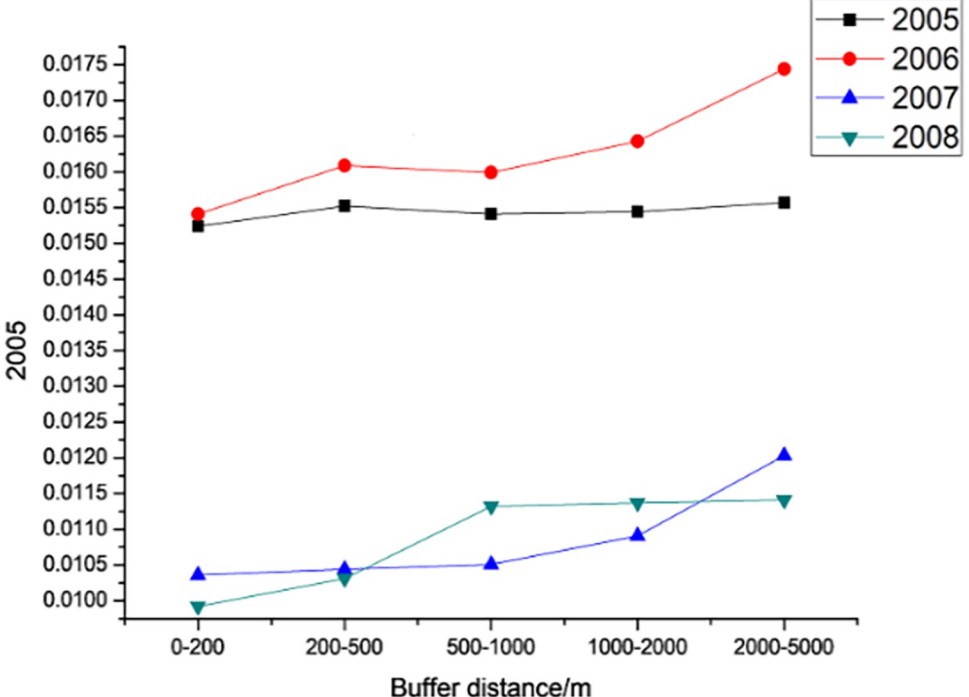

**Fig 14. Changes in VSWI by buffer zone of Dongyong Expressway from 2014 to 2016.**

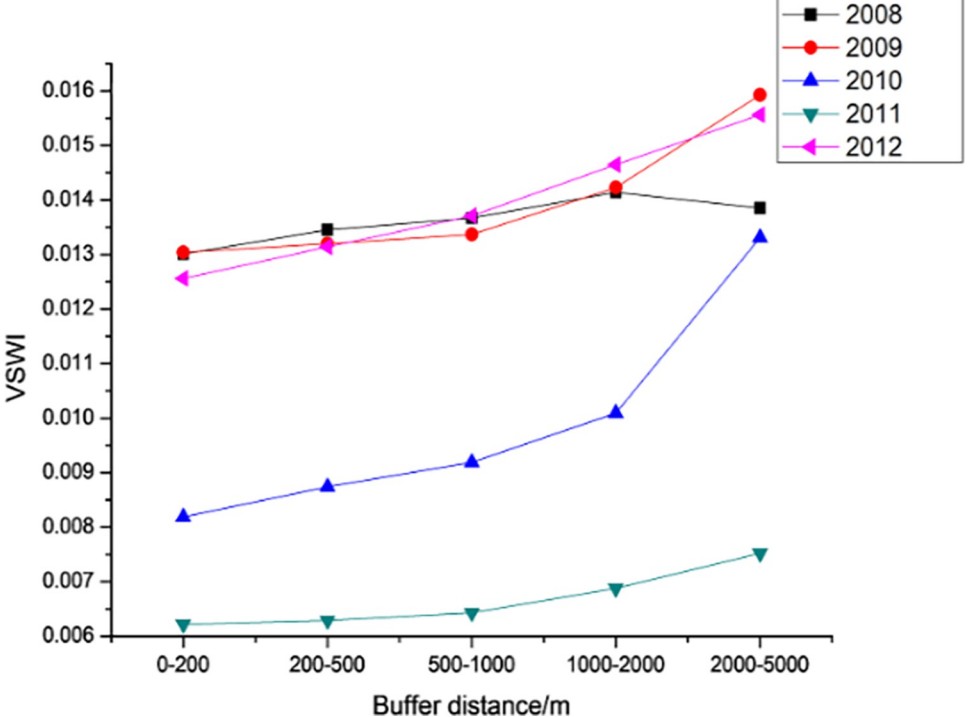

**Fig 15. Changes in VSWI by buffer zone of the Jinliwin Expressway from 2008 to 2012.**

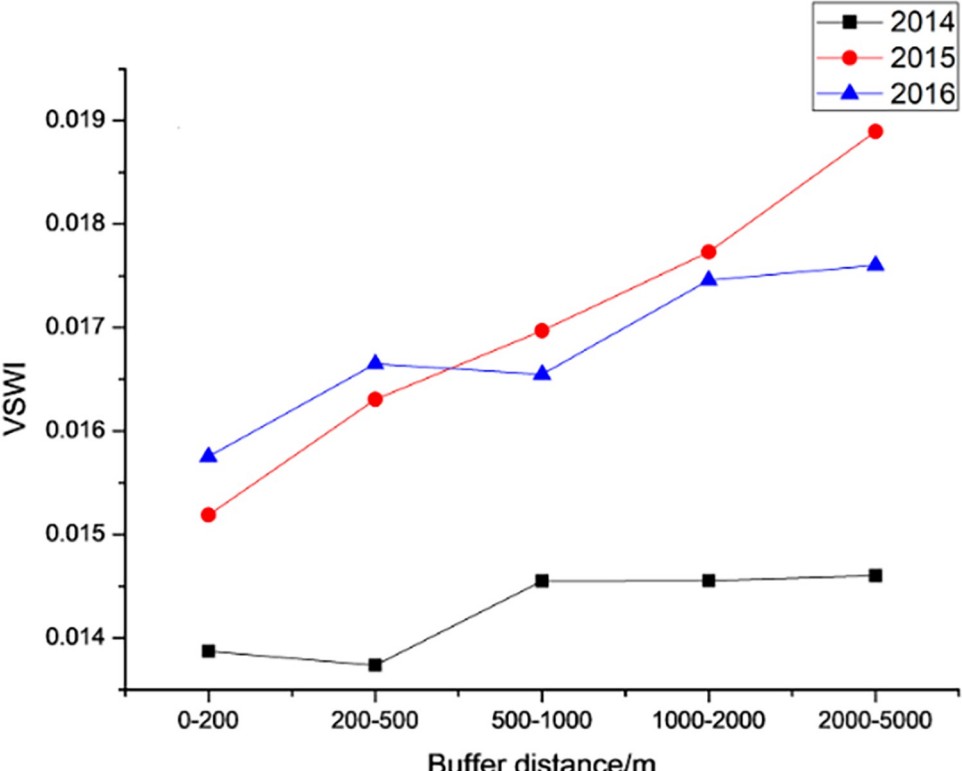

**Fig 16. Changes in VSWI by buffer zone of Yongjin Expressway from 2005 to 2008.**

Dongyong Expressway, where the average value of VSWI decreases by 44.1%, and the least decrease in VSWI is interchange node 1 on the Yongjin Expressway, which only decreased by 3.2%. In addition, it can also be seen that three interchange nodes in the high-speed section of the Jinli-Wen Expressway had a small decrease in VSWI at the time of completion but showed a significant decrease one year after completion. At the same time, the average values of the VSWI at two interchange nodes of the Dongyong Expressway showed a slight recovery one year after completion, and there were signs of recovery. This indicates that the construction of interchanges likewise has a large impact on the VSWI of road areas over time.

**3.4.2. Spatial variation pattern.** Tables 4–6 show the average values of the VSWI of each expressway interchange node at different radii before and after completion. From an overall perspective, the VSWI of the 12 interchange nodes on the three expressways shows a

**Table 3. Annual average VSWI within the 1km radius of each interchange node before and after its construction (Show only part, S1 Table for complete table).**

| node | state | Last year | During construction | Finished |
|---|---|---|---|---|
| 1 | Year | 2005 | 2006 | 2007 |
| | Value | 0.02082 | 0.02016 | 0.01987 |
| 3 | Year | 2008 | 2009 | 2010 |
| | Value | 0.01263 | 0.01203 | 0.00964 |
| 9 | Year | 2014 | 2015 | 2016 |
| | Value | 0.01903 | 0.01063 | 0.01572 |

**Table 4. Annual average VSWI within the radius of each interchange node of Yongjin Expressway before and after its construction(Show only part, S2 Table for complete table).**

| Point | Buffer distance/m | 2005 | 2006 | 2007 |
|---|---|---|---|---|
| 1 | 0–200 | 0.01664 | 0.01721 | 0.01682 |
| | 200–500 | 0.02012 | 0.01956 | 0.01777 |
| | 500–1000 | 0.02082 | 0.02016 | 0.01987 |
| | 1000–2000 | 0.02122 | 0.02292 | 0.02036 |
| | 2000–5000 | 0.01971 | 0.02312 | 0.02281 |

**Table 5. Annual average VSWI within the radius of each interchange node before and after the construction of the Jinliwin Expressway(Show only part, S3 Table for complete table).**

| Point | Buffer distance/m | 2005 | 2006 | 2007 |
|---|---|---|---|---|
| 3 | 0–200 | 0.01249 | 0.01432 | 0.00866 |
| | 200–500 | 0.01287 | 0.01263 | 0.00879 |
| | 500–1000 | 0.01263 | 0.01203 | 0.00964 |
| | 1000–2000 | 0.01283 | 0.01246 | 0.00976 |
| | 2000–5000 | 0.01387 | 0.01493 | 0.01244 |

**Table 6. Annual average VSWI within the radius of each interchange node of Dongyong Expressway before and after its construction.**

| Point | Buffer distance/m | 2005 | 2006 | 2007 |
|---|---|---|---|---|
| 9 | 0–200 | 0.01461 | 0.00865 | 0.01488 |
| | 200–500 | 0.01669 | 0.00912 | 0.01619 |
| | 500–1000 | 0.01903 | 0.01063 | 0.01572 |
| | 1000–2000 | 0.02089 | 0.01097 | 0.01786 |
| | 2000–5000 | 0.02243 | 0.01581 | 0.01929 |
| 12 | 0–200 | 0.01077 | 0.01092 | 0.01366 |
| | 200–500 | 0.01256 | 0.01205 | 0.01402 |
| | 500–1000 | 0.01339 | 0.01272 | 0.01452 |
| | 1000–2000 | 0.01441 | 0.01514 | 0.0173 |
| | 2000–5000 | 0.01491 | 0.0163 | 0.01815 |

significant increasing trend with an increase in the road median distance from the distance, which is consistent with the theory that "the farther the distance from the road, the better the VSWI trend". In general, the average VSWI is the smallest in the radiation radius of 0-200m, and the VSWI in the radiation radius of 2–5 km has a significant increase compared to the average VSWI at a radiation radius of 0-200m. Second, in terms of cross-sectional time, all interchange nodes have a decreasing trend of VSWI averages within each radiation radius after construction, which indicates that large-scale interchange construction has a significant spatial impact on VSWI; however, one year after construction, VSWI averages within some radiation radii rebound slightly.

Table 4 shows the average values of the VSWI of seven interchange nodes on the Yongjin Expressway in different years within different radiation radii before, at the beginning, and one year after the completion of the construction. As can be seen from the graphs, the VSWI averages of the interchange nodes 7 and 8 on the Yongjin Expressway in 2005 show a gradual decrease with the increase of radiation radius, which may be due to the fact that the two nodes are in mountainous locations and close to each other, which in turn generates data errors.

From the longitudinal space, the average value of the VSWI at each interchange node increases gradually with an increase in the radiation radius. With the increase in construction year, the VSWI again shows a decreasing trend.

Table 5 shows the VSWI averages of the three interchange nodes on the high-speed section of the Jinliwine Expressway for different years in different radiation radii before, at the beginning, and one year after the completion of construction. From the longitudinal space, the average VSWI of each node is in an increasing state with the increase of radiation radius, and the average VSWI of node 10 has a sudden increase between 0.5–1 km and 1–2 km of radiation radius. Horizontally, all the interchange nodes had no significant change in the mean VSWI value at the beginning of construction, but there was a large decrease in the mean VSWI value one year after construction. In general, the VSWI of all interchange nodes increases with distance within a 6 km radius of radiation.

Table 6 shows the VSWI averages of the two interchange nodes on the Dongyong Expressway for different years within different radii of radiation before, at the beginning and one year after completion. From the longitudinal space, the average VSWI of each node is in an increasing state with the increase of radiation radius, among which the average VSWI of node 9 has a larger increase between 0.5–1 km and 1–2 km radiation radius. Horizontally, there is a significant decrease in the average VSWI of all interchange nodes at the beginning of the construction, but there is a bound in the average VSWI one year after construction.

Table 7 shows the average values of VSWI for radii of 2, 4, 6, 8, and 10 km radiated by the Yongjin Expressway interchange nodes before and after construction. All interchange nodes have a significant decreasing trend of the VSWI average after construction, but one year after construction, the average VSWI rebounded slightly. In general, the VSWI of all interchange nodes increased with distance within a radius of 8 km of radiation.

From Table 7, it can be seen that the VSWI values within a 0–2 km radius of the interchange node have the greatest variation, which shows that the 0–2 km buffer zone is the most obvious area affected by the newly established interchange. From the 0–2 km buffer zone to the 6–8 km buffer zone, the VSWI shows an increasing trend, which indicates that the further away from the interchange node, the smaller the decrease in VSWI. After 8 km, the VSWI value starts to increase with time, which shows that the impact of the new interchange on VSWI is approximately 8 km.

Fig 17 shows the interannual VSWI change rates of each interchange node in the buffer zone and its corresponding high-speed before and after the new construction, where the interannual VSWI change rates of interchange nodes 1, 2, 3, 4, 8, 9, and 10 are smaller than the interannual VSWI change rates of their corresponding high-speed, and the interannual VSWI change rates of interchange nodes 5, 6, 7, 11, and 12 are larger than the interannual VSWI change rates of their corresponding high-speed. The average interannual variation rate of the interchange nodes is -0.44%, and the average interannual variation rate of their corresponding high speeds is -0.33%, which shows that the impacts of the newly established interchange nodes and high speeds on the road domain VSWI are similar.

**Table 7. Annual average VSWI values in different radiation radii of each interchange node of Yongjin Expressway before and after its construction(Show only part, S4 Table for complete table).**

| Point | Buffer distance/km | 2005 | 2006 | 2007 | 2008 | 2009 |
|---|---|---|---|---|---|---|
| 1 | 0–2 | 0.02122 | 0.02292 | 0.02036 | 0.01934 | 0.01837 |
| | 2–4 | 0.01931 | 0.01843 | 0.01821 | 0.01730 | 0.01643 |
| | 4–6 | 0.01969 | 0.01902 | 0.01846 | 0.01754 | 0.01666 |
| | 6–8 | 0.02128 | 0.01960 | 0.01879 | 0.01785 | 0.01696 |
| | 8–10 | 0.02015 | 0.20990 | 0.02101 | 0.02143 | 0.02186 |

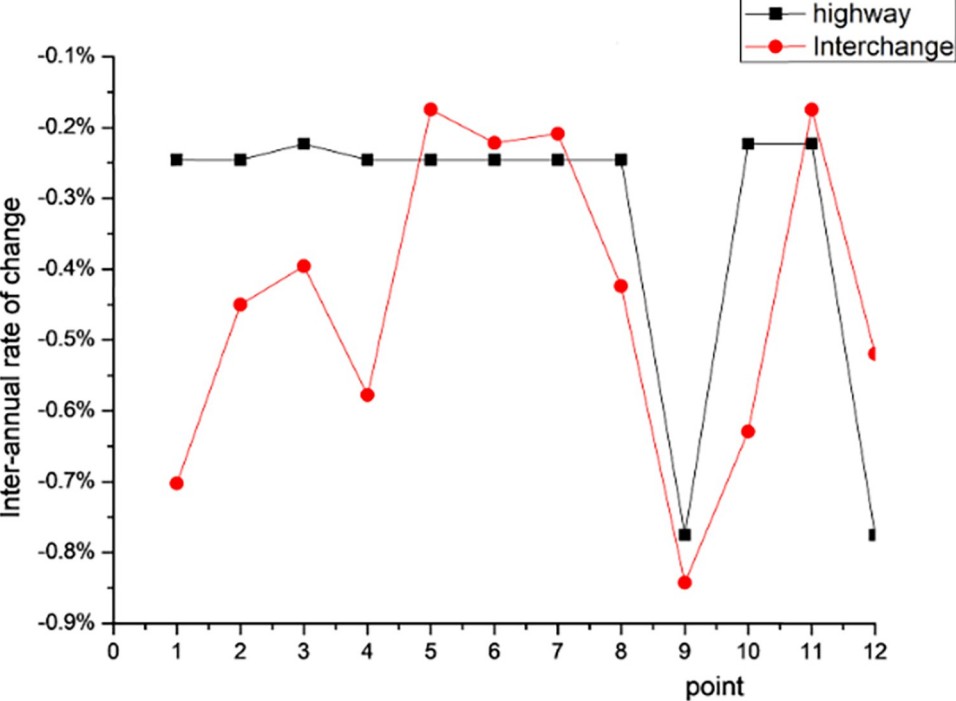

**Fig 17. Interannual VSWI change rate comparison.**

In this study, it was found that the mean value of VSWI within the 1-km buffer zone decreased significantly after the completion of the expressway, and increased with the increase of the buffer distance. Similarly, the mean value of VSWI within a 10-km radius of the interchange node showed a significant decreasing trend after the completion of the expressway. The VSWI within the 0–2 km radius of the interchange node showed the greatest change, indicating that the 0–2 km buffer zone is the area most affected by the new interchange. This is caused by human disturbance, and the conversion of natural land into highway land directly affects the plant cover, thus reducing the water retention capacity of the ecosystem and affecting the environmental humidity along the highway. The trend analysis revealed that the new highway had a significant impact on the VSWI of the road area for at least two years, and the permanent occupation of land by the highway destroyed all vegetation in the road area. Therefore, during the construction and subsequent road operation, attention should be paid to the revegetation of the road area so that the damaged ecological environment can be restored to the maximum extent. Spatially, most highways generally show the state of "the further away from the road, the higher the VSWI", and the impact of new overpasses on VSWI is about 8 km. The construction of highways and overpasses influenced the changes of VSWI in the buffer zone; the intensity of human activities affecting VSWI was reduced in places farther away from the road.

Owing to the many types of natural and anthropogenic factors affecting VSWI and the complex internal relationships, coupled with the diversity and difficulty in quantifying human activities, there is still a great deal of uncertainty in judging the mutual effects of natural and anthropogenic factors on the spatial and temporal changes in VSWI. In this study, considering the rapid development of road construction in the study area in recent years, the spatial and temporal changes of VSWI in the study area were only explored in terms of land use changes caused by road construction in human activities. Meteorology and topography are also

important factors affecting changes in VSWI; future studies should take these factors into consideration for in-depth exploration.

## 4. Conclusion

In this study, the vegetation supply water index (VSWI) of central Zhejiang region from 2005 to 2016 was extracted quantitatively by using landsat7 satellite remote sensing data, using spatial analysis method and linear trend fitting method, combining two main means of buffer zone analysis and trend fitting method, and analyzing the effect of new expressway and interchange nodes on the vegetation supply water index (VSWI) generated The spatial and temporal divergence patterns were analyzed, and the following main conclusions were drawn.

### 4.1. Overall trend of VSWI

The vegetation supply water index (VSWI) in the study area has a 12-year mean value of 0.01879, ranging from to 0.01035–0.02774, with a flat decreasing trend in the annual mean VSWI. The study area was mostly in the tertiary area, with an area share of 63.7%. According to the trend analysis method, 78.15% of the regions show a negative growth trend, with a mean interannual change rate of -1.006%, and the VSWI shows a flat decreasing trend in general.

### 4.2. Impact of new expressways on VSWI

The average value of VSWI within 1 km of the buffer zone for most expressways decreased significantly after completion, and the average value of VSWI increased as the buffer zone distance increased, indicating that the impact of new expressways on VSWI was significant.

Through the method of trend analysis, the analysis of the impact on VSWI before and after the construction of the three expressways shows that the VSWI in the buffer zone area after the completion of the Dongyong Expressway has a continuous decreasing trend, and the VSWI in the high-speed sections of the Yongjin Expressway and Jinliwin Expressway has a fluctuating decreasing trend after completion, but it has not returned to the level before the completion, which shows that the impact of the new expressway on the VSWI in its buffer zone will continue for many years, and it is clear that the impact of new expressways on the VSWI in the buffer zone will last for many years and produce a certain degree of irreversible decline. Spatially, the VSWI is generally farther from the expressway.

### 4.3. The impact of the newly established intersection on VSWI

After the completion of the new interchange, the average VSWI of the interchange nodes with a 1 km radius is in a decreasing state. From an overall perspective, the VSWI of the 12 interchange nodes on the three expressways shows a significant increasing trend with an increase in the road median distance from the distance, which is consistent with the theory that "the farther the distance from the road, the better the VSWI trend". The average value of VSWI within the radius of all interchange nodes decreases after construction, which indicates that large-scale interchange construction has a significant impact on VSWI in space, but the average value of VSWI within the radius of some interchanges slightly increases again one year after construction. In the buffer zone of each new high-speed interchange node, the buffer zone of 0–2 km is the most obvious area affected by the new interchange, and the further away from the interchange node, the smaller the decrease in VSWI. After 8 km, the VSWI value starts to increase again with time, which shows that the impact of the new interchange on VSWI is approximately 8 km.

According to the interannual VSWI change rate of each interchange node in the buffer zone and its corresponding expressway before and after the new construction, the average annual change rate of the interchange node is-0.44%, and the average annual change rate of its corresponding expressway is -0.33%, which shows that the impacts of the new interchange node and expressway on the VSWI of the road area are similar.

The construction of the highway has affected the regional ecological environment along the route to a certain extent. Therefore, in the planning and design stage, existing roads and reconstructed sections should be fully utilized to consider the protection of natural landscape, avoiding ecologically sensitive areas such as wetlands and natural services, and minimizing the impact on the ecosystem. In addition, the construction of ecological road networks should be strengthened to reduce ecological risks, such as the division and dynamic interference of highway construction on the landscape, and the design of effective ecological corridors to eliminate the isolation effect of highways. Vegetation restoration and landscape protection, biodiversity protection and ecosystem stability maintenance should be considered in the construction and operation of highways.

## Supporting information

**S1 Table. Annual average VSWI within the 1km radius of each interchange node before and after its construction.**
(DOCX)

**S2 Table. Annual average VSWI within the radius of each interchange node of Yongjin Expressway before and after its construction.**
(DOCX)

**S3 Table. Annual average VSWI within the radius of each interchange node before and after the construction of the Jinliwin Expressway.**
(DOCX)

**S4 Table. Annual average VSWI values in different radiation radii of each interchange node of Yongjin Expressway before and after its construction.**
(DOCX)

## Author Contributions

**Conceptualization:** Yongyi Li.

**Formal analysis:** Zhihao Li, Zexuan Jiao.

**Funding acquisition:** Xingli Jia.

**Investigation:** Yongyi Li.

**Methodology:** Yongyi Li.

**Project administration:** Xingli Jia.

**Resources:** Zexuan Jiao.

**Software:** Yongyi Li.

**Supervision:** Yongyi Li.

**Validation:** Zhan Xiao.

**Visualization:** Yongyi Li.

**Writing – original draft:** Yongyi Li.

**Writing – review & editing:** Yongyi Li.

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
