## [Decision Letter · Decision Letter 0]

26 Dec 2022

PONE-D-22-27887Analysis of the impact of expressway construction on soil moisture in road areasPLOS ONE

Dear Dr. Li

Thank you for submitting your manuscript to PLOS ONE. After careful consideration, we feel that it has merit but does not fully meet PLOS ONE’s publication criteria as it currently stands. Therefore, we invite you to submit a revised version of the manuscript that addresses the points raised during the review process.

We look forward to receiving your revised manuscript.

Kind regards,

Tunira Bhadauria, Ph.D.

Academic Editor

PLOS ONE

Journal Requirements:

2. Please note that PLOS ONE has specific guidelines on code sharing for submissions in which author-generated code underpins the findings in the manuscript. In these cases, all author-generated code must be made available without restrictions upon publication of the work. Please review our guidelines at https://journals.plos.org/plosone/s/materials-and-software-sharing#loc-sharing-code and ensure that your code is shared in a way that follows best practice and facilitates reproducibility and reuse. New software must comply with the Open Source Definition.

3. We note that Figures 2, 3, 4, 5, 8, 9 and 10 in your submission contain map images which may be copyrighted. All PLOS content is published under the Creative Commons Attribution License (CC BY 4.0), which means that the manuscript, images, and Supporting Information files will be freely available online, and any third party is permitted to access, download, copy, distribute, and use these materials in any way, even commercially, with proper attribution. For these reasons, we cannot publish previously copyrighted maps or satellite images created using proprietary data, such as Google software (Google Maps, Street View, and Earth). For more information, see our copyright guidelines: http://journals.plos.org/plosone/s/licenses-and-copyright.

      1. You may seek permission from the original copyright holder of Figure(s) [#] to publish the content specifically under the CC BY 4.0 license. 

“Natural Science Foundation of Shaanxi Province, No. 2020JM-260

National Key Research and Development Program of China, No. 2020YFC15120003

Shanxi Provincial Key Research and Development Project, No. 2021SF-514

This research was funded by Natural Science Foundation of Shaanxi Province, grant number No. 2020JM-260, and National Key Research and Development Program of China, grant number No. 2020YFC15120003, and Shanxi Provincial Key Research and Development Project, grant number No. 2021SF-514.”

Reviewers' comments:

Reviewer's Responses to Questions

**Comments to the Author**

1. Is the manuscript technically sound, and do the data support the conclusions?

Reviewer #1: Yes

Reviewer #2: Yes

2. Has the statistical analysis been performed appropriately and rigorously? 

Reviewer #1: Yes

Reviewer #2: Yes

3. Have the authors made all data underlying the findings in their manuscript fully available?

Reviewer #1: Yes

Reviewer #2: Yes

4. Is the manuscript presented in an intelligible fashion and written in standard English?

Reviewer #1: Yes

Reviewer #2: Yes

5. Review Comments to the Author

Reviewer #1: After carefully reading, I found that the study has its own merits in the scenario of climate changes due to developmental phenomenon by compromising with the environments and ecosystems. However, few minor corrections are required.

1. The figure 1 in introduction is not clear.

2. Introduction is in the preliminary form; kindly rewrite it in most relevant way that can express the important of study, I mean why the study was required? If not conducting research, what would be the negative impact on environment and ecosystem citing relevant references?

3. The geographic remote sensing data in this study were obtained from Landsat7 data products, is it free of cost or permission is essential before conducting research based on the data? If yes, the permission has been taken?

4. Please check the Short forms, write in full form atleast once.

5. Figure 5 is not clear, increase the visibility.

6. Most of the figures are irrelevant that can be submitted as supplementary figure, especially those satellite images.

7. Check the resolution of all the pictures.

8. Tables must be adjusted as supplementary table.

9. Check for grammar, spelling mistakes and[ .,].

Reviewer #2: Manuscript are recommended for publication with following comments for improvement

1.In manuscript at least for first time author must write full form of VSWI in abstract

2.Discussion are very short and very poor. Author are suggested to improve discussion.

3.Discussion are looking like results pl rewrite it

6. PLOS authors have the option to publish the peer review history of their article (what does this mean?). If published, this will include your full peer review and any attached files.

Reviewer #1: No

Reviewer #2: No

<quillbot-extension-portal></quillbot-extension-portal>

---

## [Author Response · Author response to Decision Letter 0]

19 Feb 2023

Original Manuscript ID: Access-2022-05031 

Original Article Title: “Analysis of the impact of expressway construction on soil moisture in road areas”

To: Plos One Editor

Re: Response to reviewers

Dear Editor,

Thank you for allowing a resubmission of our manuscript, with an opportunity to address the reviewers’ comments.

After careful consideration of the review comments of the experts, we will reply to the relevant questions as follows:

Editor#, Concern # 1: 

Comments: Please ensure that your manuscript meets PLOS ONE's style requirements, including those for file naming.

Comments: I modified the manuscript format according to the template of PLOS ONE.

Editor#, Concern # 2: 

Comments: Please note that PLOS ONE has specific guidelines on code sharing for submissions in which author-generated code underpins the findings in the manuscript. In these cases, all author-generated code must be made available without restrictions upon publication of the work. Please review our guidelines at https://journals.plos.org/plosone/s/materials-and-software-sharing#loc-sharing-code and ensure that your code is shared in a way that follows best practice and facilitates reproducibility and reuse. New software must comply with the Open Source Definition.

Comments: This manuscript uses two softwares ARCGIS and ENVI for statistical analysis and data processing. No new development program is involved and no code needs to be shared.

Editor#, Concern # 3: 

Comments: We note that Figures 2, 3, 4, 5, 8, 9 and 10 in your submission contain map images which may be copyrighted. All PLOS content is published under the Creative Commons Attribution License (CC BY 4.0), which means that the manuscript, images, and Supporting Information files will be freely available online, and any third party is permitted to access, download, copy, distribute, and use these materials in any way, even commercially, with proper attribution. For these reasons, we cannot publish previously copyrighted maps or satellite images created using proprietary data, such as Google software (Google Maps, Street View, and Earth).

Comments: According to the editor's opinion, we have re-checked figures 2, 3, 4, 5, 8, 9 and 10 of the original manuscript. The basic data of this study (including administrative divisions, satellite remote sensing data and road information) were obtained from geospatial data Cloud and Data Center for Resources and Environmental Sciences, Chinese Academy of Sciences. All the basic data were publicly available and free of charge. All images were analyzed and produced by the author using ARCGIS and ENVI software. They are not reproduced or copied, and there is no copyright problem.

Editor#, Concern #4: 

Comments: Please state what role the funders took in the study. If the funders had no role, please state: "The funders had no role in study design, data collection and analysis, decision to publish, or preparation of the manuscript."

Comments: Based on editorial advice, we reviewed our financial disclosures. We have modified the project funding of the Fund:

1. National Key Research and Development Program of China 2021YFB2600403. 2. National Key Research and Development Program of China 2020YFC1512003. 3. Fundamental Research Funds for the Central Universities, CHD 300102212203.

This research was funded by National Key Research and Development Program of China , grant number No. 2021YFB2600403, and National Key Research and Development Program of China, grant number No. 2020YFC15120003, and Fundamental Research Funds for the Central Universities, grant number No. 300102212203.

The funders had no role in study design, data collection and analysis, decision to publish, or preparation of the manuscript. It is also attached to the cover letter.

Editor#, Concern # 5: 

Comments: In your Data Availability statement, you have not specified where the minimal data set underlying the results described in your manuscript can be found. PLOS defines a study's minimal data set as the underlying data used to reach the conclusions drawn in the manuscript and any additional data required to replicate the reported study findings in their entirety. All PLOS journals require that the minimal data set be made fully available.

Comments: In response to editorial comments, we have resubmitted the minimum data set as part of the manuscript reupload process.

Reviewer#1, Concern # 1: 

Comments: The figure 1 in introduction is not clear.

Author response: Based on expert advice, the resolution of Figure 1 in the introduction has been adjusted to make it clearly visible

Reviewer#1, Concern # 2: 

Comments: Introduction is in the preliminary form; kindly rewrite it in most relevant way that can express the important of study, I mean why the study was required? If not conducting research, what would be the negative impact on environment and ecosystem citing relevant references?

Author response: The introduction section has been rewritten based on expert opinion. The significance of the study and the importance of the study were emphasized.

“As a product of infrastructure construction, the construction of highways is bound to have an impact on VSWI. Since roads are ribbon structures, their construction also affects the area along the road, and the road itself affects the vegetation and soil along the road. At present, most studies only discuss the surface temperature of the road network in urban areas, however, highways, as an important part of the development of social infrastructure, should pay attention to sustainable development during the sustainable development of roads and conduct in-depth research and analysis based on their own changes, and if the environmental disturbance of road construction cannot be analyzed, the sustainable environmental development along the road cannot be development is assessed, which in turn affects road network planning. In this paper, in order to discuss the changes of VSWI along the road domain caused by the continuous construction of the road and its impact on the surrounding environment, internal analysis of soil moisture in the regional road network area is conducted from the perspective of soil moisture along the road, using Landsat 7 remote sensing image data to analyze the evolution of the influence mechanism of soil moisture along the road in the process of continuous changes, and also to explore the influence factors of the road itself. It aims to guide the future road network encryption planning process in terms of VSWI and contribute to environmental protection, which is important for the sustainable and healthy development of roads.”

Reviewer#1, Concern # 3: 

Comments: The geographic remote sensing data in this study were obtained from Landsat7 data products, is it free of cost or permission is essential before conducting research based on the data? If yes, the permission has been taken?

Author response: According to the expert opinions, the sources of geographic remote sensing data used in the study have been verified, and the landsat7 remote sensing data used in the paper are from the Geospatial Data Cloud, which are all free data publicly available on the Internet.

Reviewer#1, Concern # 4: 

Comments: Please check the Short forms, write in full form at least once.

Author response: According to the experts' comments, all terminology abbreviations and table contents were checked and revised, and the complete terminology was stated at least once.

Reviewer#1, Concern # 5: 

Comments: Figure 5 is not clear, increase the visibility.

Author response: According to the experts' opinions, the resolution and images of Figure 5 were modified to represent the satellite images of 2015, and the rest of the images were placed in the supplementary information.

Reviewer#1, Concern # 6: 

Comments: Most of the figures are irrelevant that can be submitted as supplementary figure, especially those satellite images.

Author response: Based on expert opinion, the relevance of the images in the text was reviewed and some of the images were transferred to the supplementary information.

Reviewer#1, Concern # 7: 

Comments: Check the resolution of all the pictures.

Author response: The resolution of the images in the text was reviewed and corrected based on expert opinion.

Reviewer#1, Concern #8: 

Comments: Tables must be adjusted as supplementary table.

Author response: Based on expert opinion, the excessively long table in the text was adjusted to a supplementary table and only part of it is shown in the text.

Reviewer#1, Concern # 9: 

Comments: Check for grammar, spelling mistakes and[ .,].

Author response: Review the text for grammatical and spelling errors and correct errors based on expert opinion.

Reviewer#2, Concern # 1: 

Comments: In manuscript at least for first time author must write full form of VSWI in abstract

Author response: According to the experts' comments, all terminology abbreviations were checked and revised, and the complete terminology was stated at least once.

Reviewer#2, Concern # 2: 

Comments: Discussion are very short and very poor. Author are suggested to improve discussion.

Author response: The discussion section was revised according to the experts' opinions. The original “4 Discussion” section was changed into the “5 Conclusion” and modified; the original “3 Result” were modified into “3 Results and discussion” according to the format requirements, and new content was added at the end of the chapter. As shown below:

“In this study, it was found that the mean value of VSWI within the 1-km buffer zone decreased significantly after the completion of the expressway, and increased with the increase of the buffer distance. Similarly, the mean value of VSWI within a 10-km radius of the interchange node showed a significant decreasing trend after the completion of the expressway. The VSWI within the 0-2 km radius of the interchange node showed the greatest change, indicating that the 0-2 km buffer zone is the area most affected by the new interchange. This is caused by human disturbance, and the conversion of natural land into highway land directly affects the plant cover, thus reducing the water retention capacity of the ecosystem and affecting the environmental humidity along the highway. The trend analysis revealed that the new highway had a significant impact on the VSWI of the road area for at least two years, and the permanent occupation of land by the highway destroyed all vegetation in the road area. Therefore, during the construction and subsequent road operation, attention should be paid to the revegetation of the road area so that the damaged ecological environment can be restored to the maximum extent. Spatially, most highways generally show the state of "the further away from the road, the higher the VSWI", and the impact of new overpasses on VSWI is about 8 km. The construction of highways and overpasses influenced the changes of VSWI in the buffer zone; the intensity of human ac-tivities affecting VSWI was reduced in places farther away from the road.

Owing to the many types of natural and anthropogenic factors affecting VSWI and the complex internal relationships, coupled with the diversity and difficulty in quantifying human activities, there is still a great deal of uncertainty in judging the mutual effects of natural and anthropogenic factors on the spatial and temporal changes in VSWI. In this study, considering the rapid development of road construction in the study area in recent years, the spatial and temporal changes of VSWI in the study area were only explored in terms of land use changes caused by road construction in human activities. Meteorology and to-pography are also important factors affecting changes in VSWI; future studies should take these factors into consideration for in-depth exploration.”

Reviewer#2, Concern # 3: 

Comments: Discussion are looking like results pl rewrite it.

Author response: The discussion section was revised according to the experts' opinions. The original “4 Discussion” section was changed into the “5 Conclusion” and modified.

Best regards,

<Yongyi Li> et al.

---

## [Editor Report · Decision Letter 1]

6 Mar 2023

Analysis of the impact of expressway construction on soil moisture in road areas

PONE-D-22-27887R1

Dear Dr. Li

We’re pleased to inform you that your manuscript has been judged scientifically suitable for publication and will be formally accepted for publication once it meets all outstanding technical requirements.

Kind regards,

Tunira Bhadauria, Ph.D.

Academic Editor

PLOS ONE

Additional Editor Comments (optional):

Reviewers' comments:

<quillbot-extension-portal></quillbot-extension-portal>

---

## [Editor Report · Acceptance letter]

21 Mar 2023

PONE-D-22-27887R1 

Analysis of the impact of expressway construction on soil moisture in road areas 

Dear Dr. Li:

I'm pleased to inform you that your manuscript has been deemed suitable for publication in PLOS ONE. Congratulations! Your manuscript is now with our production department. 

Kind regards, 

on behalf of

Dr. Tunira Bhadauria 

Academic Editor

PLOS ONE